# Functional Restoration of Pituitary after Pituitary Allotransplantation into Hypophysectomized Rats

**DOI:** 10.3390/cells10020267

**Published:** 2021-01-29

**Authors:** Jai Ho Choi, Jung Eun Lee, Hong-Lim Kim, Seung Hyun Ko, Se Hoon Kim, Seung Ho Yang

**Affiliations:** 1Department of Neurosurgery, Seoul St. Mary’s Hospital, College of Medicine, The Catholic University of Korea, 222 Banpodaero, Seochogu, Seoul 06591, Korea; bivalvia@catholic.ac.kr; 2Cell Death Disease Research Center, Department of Neurosurgery, St. Vincent Hospital, College of Medicine, The Catholic University of Korea, 222 Banpodaero, Seochogu, Seoul 06591, Korea; eunree@nate.com; 3Integrative Research Support Center, Laboratory of Electron Microscope, College of Medicine, The Catholic University of Korea, 222 Banpodaero, Seochogu, Seoul 06591, Korea; wgwkim@catholic.ac.kr; 4Department of Endocrinology, St. Vincent Hospital, College of Medicine, The Catholic University of Korea, 222 Banpodaero, Seochogu, Seoul 06591, Korea; kosh@catholic.ac.kr; 5Department of Pathology, Yonsei University of College of Medicine, Seochogu, Seoul 06591, Korea; paxco@yuhs.ac

**Keywords:** pituitary gland, transplantation, hypophysectomy

## Abstract

Long-term hormone replacement therapy due to panhypopituitarism can lead to serious complications and thus, pituitary transplantation is considered a more desirable. We investigated functional restoration after allotransplatation of the pituitary gland. We transplanted extracted pituitary gland into the omentum of an hypophysectomized rat. Two experiments were performed: (1) to confirm the hypophysectomy was successful and (2) to assess functional restoration after pituitary transplantation. Pituitary hormone level and weight change were consecutively assessed. Electron microscopic (EM) examinations were performed to identify morphological changes at 3 days after transplantation. We confirmed that pituitary gland was properly extracted from 6 rats after sacrifice. The findings showed (1) a weight loss of more than 3% or (2) a weight change of less than 2% along with a decreased growth hormone (GH) level by more than 80% at 2 weeks post-hypophysectomy. A further four rats underwent pituitary transplantation after hypophysectomy and were compared with the previously hypophysectomized rats. All showed rapid weight gain during the two weeks after transplantation. The thyroid-stimulating hormone, prolactin, and GH levels were restored at one week post-transplantation and maintained for 10 weeks. Hypophyseal tissue architecture was maintained at 3 days after transplantation, as indicated by EM. These data suggest that a transplanted pituitary gland can survive in the omentum with concomitant partial restoration of anterior pituitary hormones.

## 1. Introduction

The endocrine system is regulated by the hypothalamic-pituitary-organ axis (HPA). The hypothalamus delivers precise signals to the pituitary gland. Various hormones are then released from the pituitary gland to stimulate and regulate target organs including the thyroid gland, adrenal gland, and gonads. Pituitary hormones are also associated with growth, lactation, and water balance. Consequently, the pituitary gland is the main regulator of the endocrine system [1]. Deterioration of pituitary functions can lead to serious complications including diabetes insipidus, hypothyroidism and hypocortisolemia [2]. Lifelong hormonal replacement therapy is needed in these patients, but significant adverse effects can develop through the use of a long-term hormonal replacement [3].

Organ transplantation remains the most effective treatment for end-stage organ failure, not only in most vascularized solid organs such as the liver, kidneys, heart, and lungs but also in non-vascularized islet cell transplantation for the treatment of diabetes mellitus, even though transplantation has some limitations including graft failure or rejection, histocompatibility, the use of immunosuppressants, and the long-term survival of implanted organs [4,5]. Although advances in knowledge and technique continue to improve the clinical outcomes of solid and tissue transplantation, clinical application to the pituitary gland has been limited [6].

Harvey Cushing inceptively attempted pituitary transplantation on a human patient in 1908 [6]. Since Cushing’s pioneering attempt, several researchers have investigated experimental pituitary transplantation [7,8,9,10]. Pituitary grafting to reveal the hypothalamic control of the pituitary gland was studied by Harris and Jacobshon in 1952 [8]. Halasz et al. suggested the term “hypophysiotrophic area” to support pituitary transplantation in 1962 [11]. The first experimental pituitary transplantation into various hypothalamic regions of hypophysectomized rats was conducted by Knigge et al. in 1962 [9]. In 1985, Tulipan et al. (reinforced by Maxwell et al. in 1998) demonstrated the restoration of pituitary hormones in plasma after pituitary transplantation into hypophysectomized rats [7,12]. However, further investigations regarding pituitary transplantation for functional restoration of pituitary hormones have rarely been performed since 1998. Pituitary transplantation can be used as a theoretical curable method for hypopituitarism to physiologically overcome the side effects of long-term hormonal replacement therapy.

In this study, we investigated the functional restoration of pituitary hormones after allotransplantation of the pituitary gland into omental pouches of hypophysectomized rats with severe combined immunodeficiency (SCID).

## 2. Materials and Methods

### 2.1. Animals

Male SCID Sprague Dawley rats weighing 160–280 g (Koatech, Pyungtaek, Korea) were used in this study. These animals were housed in a laboratory with a controlled temperature and a 12 h light-dark cycle. Food and water were provided ad libitum. All protocols regarding the use of animals were approved by the Institutional Animal Care and Use Committees (IACUC) of the Catholic University of Korea (SVH IRB 17-5). All experiments were carried out in accordance with the relevant guidelines and regulations.

### 2.2. Parapharyngeal Approach to Remove the Pituitary Gland

All SCID rats were anesthetized with gas anesthesia (3% isoflurane in 500 mL/min O_2_). Surgeries were performed as described previously [13]. Briefly, we made a skin incision on the midline of the anterior neck and entered into the floor of the cranium through the omohyoid muscle located at the lower edge of the sternohyoid muscle. After the retraction of the sternohyoid muscle, trachea, and esophagus laterally, the cranial bone was vertically drilled out at the midline and in front of the blue suture line with the bone burr under a binocular microscope. The last layer of cranial bone was broken off as a circular piece. The hole was widened until most of the underlying pituitary gland was exposed. We obtained the pituitary gland with a fine membrane using fine forceps.

### 2.3. Experiments to Determine Correct Removal of the Pituitary Gland

For assessing whether the pituitary gland was properly removed or not, 10 SCID rats were used. After removal of the pituitary gland, we sequentially checked serum levels of pituitary hormones including thyroid stimulating hormone (TSH), growth hormone (GH), prolactin, and adrenocorticotropic hormone (ACTH) using enzyme linked immunosorbent assays (ELISA) and weight change to confirm the appropriate hypophysectomy procedure had been performed. Rat serum was regularly obtained from the tail vein at 10 a.m. We examined serum hormone level before surgery and at 2, 4, 6, 8, 10, 12 and 14 weeks after surgery. Body weight was also examined before surgery and at 2, 4, 6, 8, 10, 12 and 14 weeks after surgery. After completion of the experiment schedules, rats were sacrificed, and the sellar region was examined to determine whether the hypophysectomy had been performed correctly. The cranial vault was removed and the right hemisphere was retracted. The midline skull base structure was examined with a binocular microscope to determine whether the pituitary gland remained.

### 2.4. Pituitary Gland Transplantation into the Omentum and the Follow-Up Protocol

For identifying whether the pituitary function could be restored after transplantation, another 4 SCID rats were used. All rats were anesthetized using gas anesthesia (3% isoflurane in 500 mL/min O_2_) and placed in a supine position. The pituitary gland was extracted from a sibling male rat and stored in histidine-tryptophan-ketoglutarate (HTK) solution. For the transplantation procedure, the abdominal area was shaved and sterilized using povidone iodine. A laparotomy procedure of approximately 2 cm was done and the omentum was pulled out. The pituitary gland of the donor was placed on the omentum and packed in scaffold form (Figure 1). The graft was put back into the abdominal cavity. The peritoneum and skin were closed layer-by-layer.

Serum levels of hormones, including TSH, GH, prolactin, and ACTH, from the tail vein were examined sequentially using ELISA. Hormone levels and body weight at baseline and at 2 weeks after hypophysectomy were evaluated to confirm the hypophysectomy was performed correctly. We transplanted the extracted pituitary gland from another SCID rat into the omental pouch of a hypophysectomized rat at three weeks after hypophysectomy. Serum hormone levels and body weight were obtained sequentially once every 2 weeks after pituitary transplantation. Details of the study protocol are illustrated in Figure 2. The serum hormone levels and body weights of the transplanted group were compared to those of the hypophysectomy group. 

### 2.5. Electron Microscopic and Immunohistochemical Examination after Pituitary Transplantation 

The implanted pituitary gland was harvested at 3 days after transplantation and assessed by electron microscopy (EM, JEM-1010, Jeol, Japan). Ultrastructural changes including nucleus degeneration, and damage to the mitochondria, rough endoplasmic reticulum (RER), and secretory granules (SGs) were evaluated. Details of tissue preparation are described in our previous report [13]. Briefly, the tissue was placed in a fixative, washed with phosphate buffer saline (PBS), and post-fixed in phosphate-buffered 1% osmium tetroxide. After dehydration with alcohol, the tissue was embedded in an EPON resin. Ultrathin sections (70–80 nm in thickness) were made with an ultramicrotome (Ultracut UCT, Leica, Austria) and subsequently stained with uranyl acetate and lead citrate. 

For histopathologic examination, the extracted pituitary glands were fixed in 10% formalin and embedded in paraffin. Sequential sections of the pituitary gland were cut to 4 µm thickness with a cryostat and mounted onto slides. After deparaffinization and rehydration with xylene, ethanol, and deionized H_2_O, these sections were stained with hematoxylin and eosin (H&E). In addition, special staining for detecting ACTH, GH, prolactin, and TSH was conducted. 

## 3. Results

### 3.1. Serum Hormone Levels and Body Weight Change after Appropriate Hypophysectomy

We performed hypophysectomy for a total of 10 SCID rats to identify a correct hypophysectomy. Of these ten rats, six were found to have undergone adequate pituitary gland removal (more than 70%) on microscopic examination after sacrifice and surgery (Figure 3). One rat showed 6.8% weight loss at two weeks after the hypophysectomy and died a few days later. Five out of the six rats showed (1) weight loss of more than 3% or (2) a weight change of less than 2% along with a decreased serum GH level by more than 80% at two weeks post-hypophysectomy. Details regarding hormone levels and weight change of properly hypophysectomized rats are depicted in Figure 4. Serum TSH levels gradually decreased after surgery and were not detected at 4–6 weeks after surgery. Serum GH and prolactin levels abruptly decreased two weeks postoperation. This was maintained until 6–8 weeks and disappeared at 10 weeks. Serum ACTH levels were maintained until 4–6 weeks after surgery and sharply dropped at 6–8 weeks postoperation. The body weight of adequately hypophysectomized rats showed almost no change until 14 weeks after surgery. In contrast, all rats (*n* = 4) with inadequate removal of the pituitary gland showed (1) gradual weight gain or (2) no change in serum hormone levels. 

### 3.2. Serum Hormone Levels and Body Weight Change after Transplantation

Another four SCID rats were used to assess serum hormone and body weight change after transplantation. These rats underwent pituitary transplantation into the omental pouch three weeks after hypophysectomy. Serum TSH and GH levels decreased by more than 80% compared to baseline hormone levels (less than 1.0 ng/mL in serum level) in all rats at two weeks post-hypophysectomy. In addition, all hypophysectomized rats showed no weight gain for two weeks. After pituitary transplantation, serum TSH levels were elevated up to the baseline level at one week post-transplantation and maintained until 11 weeks after transplantation. Serum GH and prolactin levels normalized at one week after transplantation but decreased thereafter over time. Serum ACTH levels gradually decreased after hypophysectomy and were not detected at eight weeks posthypophysectomy. Accordingly, pituitary transplantation did not affect the restoration of serum ACTH (Figure 5). All rats after receiving pituitary transplantation showed rapid weight gain until one week (246.4 ± 4.8 g) compared to baseline (167 ± 7.9 g) and gradually gained weight until 11 weeks (295 ± 4.5 g at 11 weeks) after transplantation. In contrast, hypophysectomized rats without transplantation (control group) showed no hormonal restoration or weight gain (Figure 6). Mean TSH levels in the transplantation group (4.302 ± 0.664 ng/mL) were significantly higher compared to the control group (0.018 ± 0.040 ng/mL, *p* = 0.001) at one week after transplantation and maintained until 11 weeks after transplantation (4.730 ± 0.991 versus 0 ng/mL at 11 weeks, *p* < 0.001). Serum GH (14.616 ± 6.176 versus 0.664 ± 0.426 ng/mL, *p* = 0.001) and prolactin (10.405 ± 1.786 versus 0.13 ± 0.241 ng/mL, *p* < 0.001) levels were significantly different between the two groups at one week post-transplantation but the difference gradually decreased. Mean serum GH levels were higher in the transplantation group until 11 weeks, but there was no significant difference from three weeks (7.715 ± 9.923 versus 0.696 ± 0.693 ng/mL, *p* = 0.152) after transplantation. The statistical difference of serum prolactin levels was maintained until seven weeks (4.765 ± 0.544 versus 0 ng/mL, *p* < 0.001) after transplantation. Serum ACTH levels did not show a statistical difference between the two groups. Body weight was significantly different between the two groups from five weeks (276.875 ± 8.035 versus 214.4 ± 43.281 g, *p* = 0.034) after transplantation and was maintained until 11 weeks (295 ± 4.545 versus 220.5 ± 29.894 g at 11 weeks, *p* = 0.023) after transplantation (Figure 6).

### 3.3. Immunohistochemical Staining Results and Ultrastructural Changes on Electron Microscopic Examination of Transplanted Pituitary Glands

The harvested pituitary glands from the omental pouches at three days after transplantation maintained nearly normal ultrastructure on EM examination. Structures of mitochondria, RER, and nuclei of harvested pituitary glands were maintained similarly to those of the normal gland. The number and structure of SG were also retained. However, small necrotic areas were found in the harvested pituitary glands. In addition, immunohistochemical staining was done using harvested pituitary glands. We found well-preserved adeno- and neuro-hypophysis in the H&E stain. Normal glandular structures with a normal number of acidophils and basophils and without ischemic foci were seen in anterior hypophysis. Pituitary cells secreting ACTH, GH and prolactin showed strong positive staining, but TSH showed relatively weak staining in immune staining (Figure 7)

## 4. Discussion

The present study revealed that adequately hypophysectomized rats showed weight loss or little weight gain along with decreased serum TSH and GH levels over time. In contrast, rats with inadequate removal of the hypophysis gained weight based on a normal growth curve. Previous studies have also suggested that no weight gain is an important parameter for the correct hypophysectomy of rats [10,14,15]. In this study, we additionally inspected the sellar region by binocular microscopy to confirm whether the pituitary gland was properly removed after rat sacrifice. In rats with no weight gain or minimal weight gain, more than 70% of the total volume of the pituitary gland was removed. Our results verify that weight loss or minimal weight gain is a very important sign of adequate hypophysectomy. In this study, total hypophysectomy was impossible because rats with total resection of hypophysis showed severe weight loss (more than 6% of baseline weight) and quickly died after surgery. In this study, persistent circulating ACTH for more than 4 weeks after hypophysectomy was found, even though the half-life of pituitary hormones is short. This may be due to a substantial remaining portion of the pituitary gland. However, other hormones (TSH, GH, and prolactin) rapidly decreased at 2–4 weeks after hypophysectomy and this could be due to the functional decline of the remaining pituitary tissue over time. Therefore, we performed transplantation at 3 weeks after hypophysectomy to minimize the influence of remaining pituitary tissue on the restoration of hormones after pituitary implantation.

This study also identified that some hormone levels were restored, and body weight was gained after pituitary transplantation. In 1980, Knigge et al. demonstrated that pituitary transplantation resulted in weight gain in all hypophysectomized rats [10]. They suggested that total body weight was a parameter of implant survival. The return of adrenal weight and sexual function and testicular maintenance after transplantation were also reported in their study [10]. Tulipan et al. were the first investigators to show the restoration of serum pituitary hormone levels, including prolactin and thyroxin after hypophyseal implant [12]. Maxwell et al. then evaluated graft survival and functional restoration by radioimmunoassay for serum pituitary hormone levels and also by correlative immunohistochemistry in 1998 [7]. Their assay results revealed that serum levels of hormones including prolactin, β-endorphin, GH, TSH, and luteinizing hormone (LH) were restored in many of the rats with pituitary transplantation (45–100%), and reinforced by immunohistochemical staining of the anterior lobe of the implanted gland [7]. Our results also revealed the restitution of serum TSH, prolactin and GH levels after transplantation. This was corroborated by EM examination. We assessed ultrastructural preservation of the graft by EM at 3 days after transplantation. We have previously reported that the extracted rat pituitary gland can be preserved for 3 days in HTK solution under hypothermic conditions [13]. Therefore, we conducted EM examination to compare with the EM results of our previous study. The ultrastructure of the transplanted pituitary gland in this study was more preserved than the extracted gland preserved in the HTK solution. We additionally conducted an IHC stain to evaluate the viability of a subtype of hormone-secreting cells. Positive staining of hormone-secreting cells (ATCH, TSH, prolactin, and GH) can represent functional preservation of implanted pituitary glands in this study.

In this study, we transplanted extracted pituitary gland into the omental pouch of the abdominal cavity of rats. There is no general consensus on the optimal site of pituitary transplantation. Commonly used sites for rat pituitary graft are the kidney capsule (ectopic area), the hypophysiotropic area and the third ventricle [7,9,10,11,12,16,17,18,19,20]. The kidney capsule was regarded as the most convenient area for ectopic transplantation of the pituitary gland [16,17,18]. The ectopic transplantation induced hyperprolactinemia and nearly influenced other anterior pituitary functions in some earlier studies [21]. In addition, such induced hyperprolactinemia can lead to several complications [22,23].

A key regulator of pituitary biology is the hypothalamic–pituitary relationship connected with the specialized portal system delivering hypothalamic factors to the pituitary gland. Hypothalamic input and pituitary feedback can be essential for maintaining physiological levels and patterns of pituitary hormones. The pituitary fossa may be the most important physiological site for pituitary transplantation along with specialized blood supply. However, the orthotopic transplantation of pituitary gland in rats was technically very difficult in our preliminary experiments. Pituitary transplantation to the third ventricle has been investigated by some investigators. Knigge et al. have suggested that cerebrospinal fluid (CSF) hormonal flow might be a postulated hypothalamic–pituitary control pathway. Accordingly, vascular input from portal vessels might not be a requisite for pituitary function [10]. Maxwell et al. have supported this hypothesis by performing a provocation test with exogenous hypothalamic releasing factors on rats with pituitary implant to the third ventricle [7]. Therefore, the third ventricle can be considered a physiological and desirable site for pituitary transplantation. However, in our preliminary experiments, we could not find the implanted pituitary gland in the third ventricle after a few days, in contrast to the previous study by Maxwell et al. This might have been due to the CSF circulation.

The omental pouch is widely used for pancreatic islet transplantation [24,25]. The use of the omental pouch as a transplantation site has some advantages, as follows: (1) a larger volume of tissues can be accommodated, (2) it can be more suitable for encapsulated or islet grafts with a low degree of endocrine cells, (3) secreted hormones can be delivered to systemic circulation through hepatic portal system, and (4) the implanted graft can be harvested at the end of study [24]. We could not find revascularization of transplanted pituitary gland at 3 days after transplantation. Moreover, long-term survival of transplanted glands without revascularization is a main drawback of pituitary transplantation into the omental pouch. We tried to find implanted pituitary glands at seven days after transplantation, but remaining glands could not be found in the omental pouch. In contrast, implanted pituitary glands beneath the kidney capsule can be preserved for more than eight weeks [16]. However, some reports have suggested that the omentum shows the potential for angiogenesis to promote neovascularization in ischemic tissues [26,27,28]. These results might provide the possibility of long-term survival of non-vascularized graft such as pituitary glands or islet cells inside the omental pouch. Another limitation of the omental pouch is the absence of hypothalamic-pituitary relationship connected with specialized vascular supply. Accordingly, the implanted pituitary gland may not normally respond to stress or physiological challenges. The omental pouch might not be considered a physiological site because some pituitary hormones can be excreted by hepatic clearance. In particular, GH can be excreted by hepatic clearance, ranging from 50 to 95%. Our results showed that serum GH levels surged at one week after transplantation, but rapidly declined over two weeks. This might have been due to hepatic clearance. Serum prolactin level restored at one week after transplantation and was maintained for five weeks. In the present study, there was no induced hyperprolactinemia caused by the loss of negative feedback from the hypothalamus. The mechanism is unclear, but hepatic clearance and buffering of the hepatic portal system before entering into the systemic circulation might influence the serum prolactin levels after transplantation of pituitary glands, although the kidneys are the main site of prolactin elimination [29]. To the best of our knowledge, our study is the first investigation to implant extracted pituitary glands into the omental pouch. Because the omental pouch has several limitations, additional investigations to find optimal site for pituitary transplantation and overcome these limitations are needed. Moreover, because of the limitations of other sites including the kidney capsule, the ventricle and the sellar floor as well as the omentum, further studies regarding cell transplantation, such as stem cell or separated pituitary cells, should be conducted [30,31].

Another limitation of this study was the single time-point measurements of pulsatile pituitary hormones. For accurate and physiological assessment of pituitary hormones, multiple blood sampling to discern the basal and pulsatile hormone levels would be required. However, multiple sampling within a period of hours from the tail vein may cause considerable stress and can lead to the morbidity and mortality of rats.

## 5. Conclusions

In this study, it was found that TSH, GH and prolactin were restored one week after transplantation while ACTH was not released from the graft. In addition, the structural preservation of implanted hypophysis could be assured by EM and IHC examinations. In conclusion, this study suggests that transplanted pituitary glands could survive in the omentum with concomitant partial restoration of anterior pituitary hormones.

## Figures and Tables

**Figure 1 cells-10-00267-f001:**
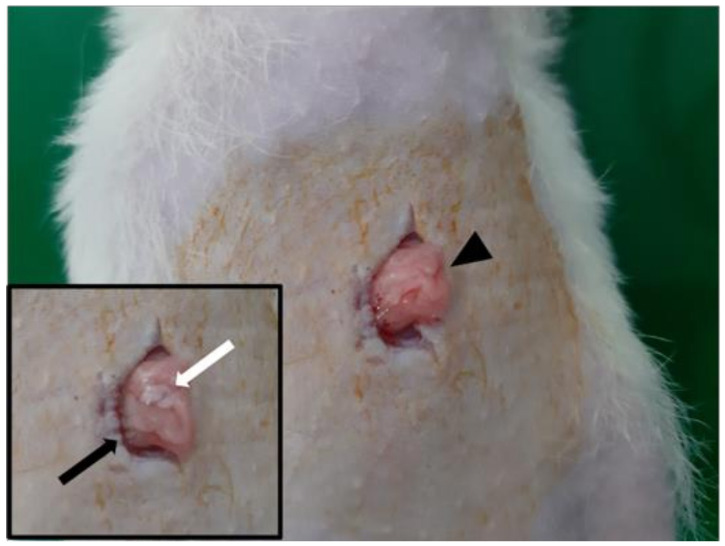
The extracted pituitary gland (white arrow) was placed on the omentum (black arrow). Thereafter, the gland was packed within the omental pouch (black arrowhead).

**Figure 2 cells-10-00267-f002:**
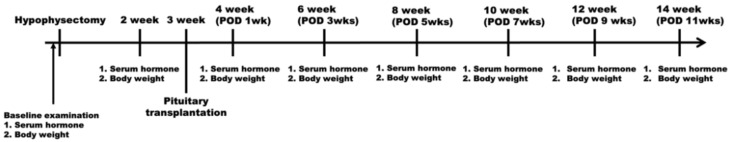
The study protocol for assessing hormonal and body weight change after pituitary transplantation into the omentum of an hypophysectomized rat.

**Figure 3 cells-10-00267-f003:**
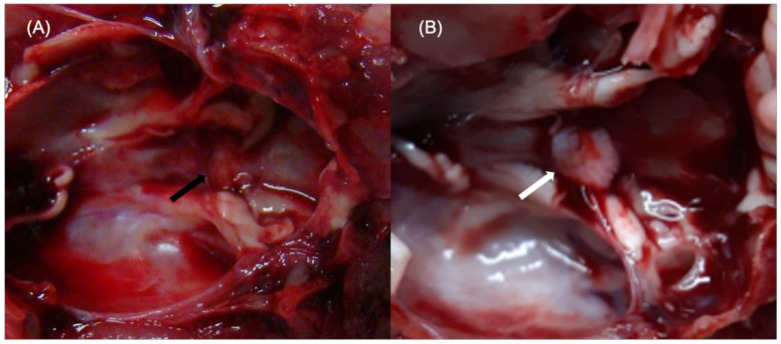
(**A**) Adequate removal of pituitary gland. A small remnant of the pituitary gland (black arrow) was shown in the sellar fossa on microscopic examination. (**B**) In contrast, large amounts of remnant pituitary gland (white arrow) were shown in cases with inadequate removal of the pituitary gland.

**Figure 4 cells-10-00267-f004:**
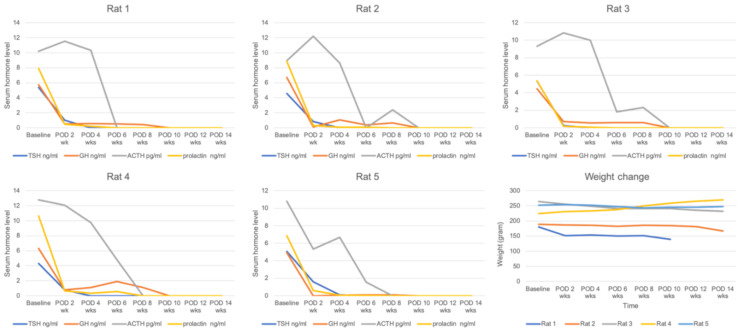
Serum hormone changes of five rats with adequate removal of the pituitary gland over time. The serum TSH level gradually decreased. It was not detected at 4–6 weeks postoperation. Serum GH and prolactin levels abruptly decreased at two weeks postoperation and were maintained until 6–8 weeks. Serum ACTH levels were maintained until 4–6 weeks and sharply dropped 6–8 weeks postoperation. The body weights of correctly hypophysectomized rats seldom changed until 14 weeks. POD, postoperative day.

**Figure 5 cells-10-00267-f005:**
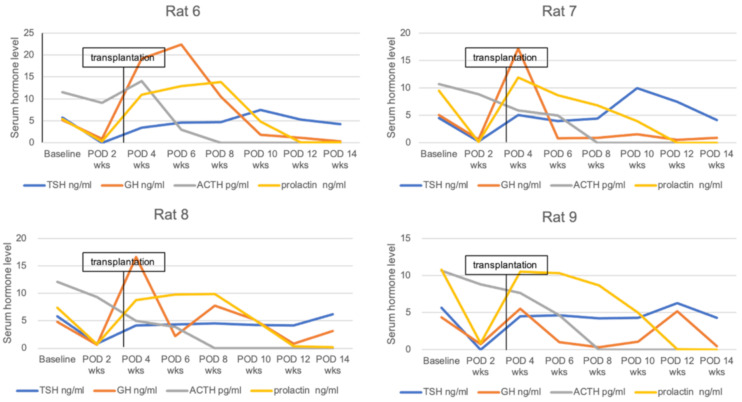
Serum hormone changes of four rats after hypophysectomy and pituitary transplantation. The serum TSH level was elevated up to the baseline level at post-transplantation 1 week and maintained until 11 weeks after transplantation. Serum GH and prolactin levels normalized at 1 week after transplantation but decreased over time. Serum ACTH level gradually decreased after hypophysectomy and was not detected at 14 weeks post-hypophysectomy. POD, postoperative day.

**Figure 6 cells-10-00267-f006:**
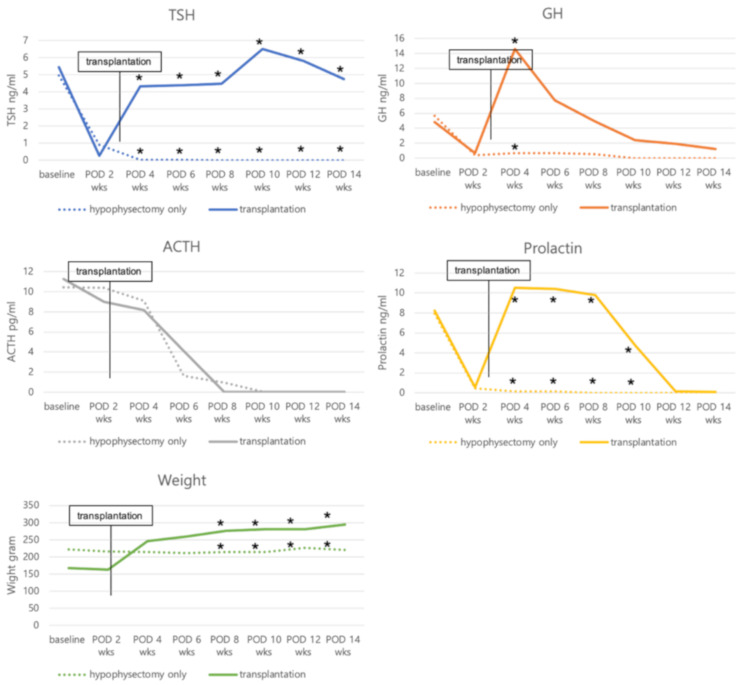
Mean TSH levels after transplantation were significantly higher compared to the hypophysectomy-only group and maintained until 11 weeks after transplantation. Serum GH and prolatin levels normalized at 1 week after transplantation and mean GH levels were significantly higher in the transplantation group than in the hypophysectomy-only group. There was no significant difference in the mean ACTH levels between the two groups. Mean body weight was significantly different between the two groups from 8 weeks posthypophysectomy. POD, postoperative day; * *p* < 0.05.

**Figure 7 cells-10-00267-f007:**
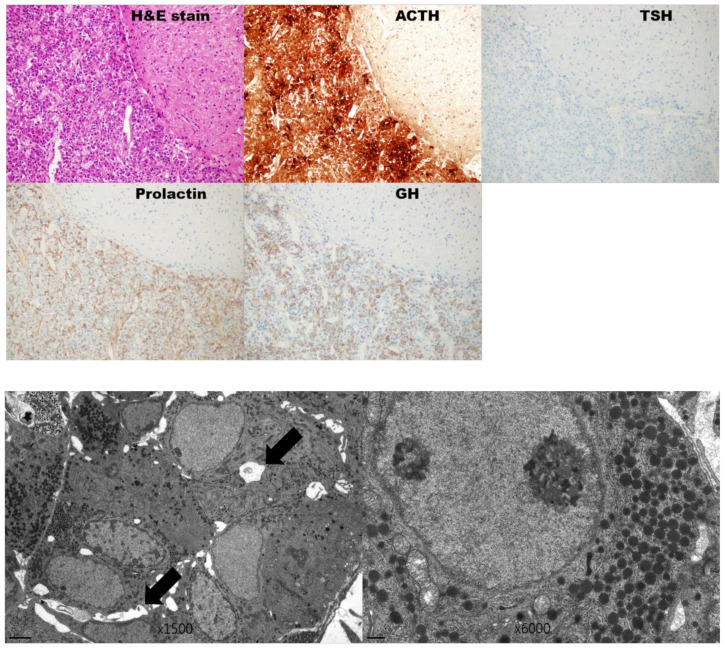
H&E stain showing well-preserved adeno- and neuro-hypophysis at 3 days after transplantation. Special immunohistochemical study showing strong positive staining for ACTH, prolactin, and GH. However, TSH showing relatively weak staining on ImmunoHistoChemistry IHC examination. (Original magnification ×200). EM examination of the harvested pituitary gland from the omental pouch at 3 days after transplantation. EM examination showed nearly normal ultrastructure except small necrotic areas (black arrow). (original magnification ×1500) Structures of mitochondria, RER, and nucleus were maintained similar to those of the normal gland. The number and structure of SGs were also retained. (original magnification ×6000).

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
