# Peer review of "Functional Restoration of Pituitary after Pituitary Allotransplantation into Hypophysectomized Rats"

_cells, 2021, doi:10.3390/cells10020267_

Round 1

Reviewer 1 Report

The authors have provided sufficient responses to my comments. I have no more requests.

Author Response

Response to Reviewer 1 Comments

Comment #1

The authors have provided sufficient responses to my comments. I have no more requests.

Response #1

Thank you very much for your comments.

Reviewer 2 Report

In the paper presented by the group of Yang, SH, et al. entitled “Functional of pituitary after pituitary allotransplantation into the hypophysectomized rats” the authors has carried out a study in which have transplanted pituitary gland in the omentum.

In my opinion, despite the impeccable work done by the authors, this study has enough gaps and difficulties to be published and would need changes in its orientation.

Specifically, the introduction did not speak of the possible problems of transplants and did not assess the current state of the problem of pituitary transplants. The results presented are limited to measuring hormones, and immunohistochemical sections. Without other references.

The conclusions are very summarized and little elaborated and, the same happens with the discussion. The references, although it is based on old works, should refer to more current publications.

The figures are in a very small format and are difficult to see.

On the other hand, the pdf that has been sent to me to evaluate looks like a draft of the manuscript. I believe that the authors, should take care of the presentation so that the reviewer does not have difficulties to read the manuscript.

Author Response

Response to Reviewer 2 Comments

Comment #1

Specifically, the introduction did not speak of the possible problems of transplants and did not assess the current state of the problem of pituitary transplants.

Response #1

Thank you for your comments. As per your recommendation, we have added the possible problems of transplantation including hypophyseal implant in the revised manuscript. All changes are highlight in red.

  1. Page 2, line 45-53: We have added “Organ transplantation remains the most effective treatment for end-stage organ failure not only in most vascularized solid organs, such as liver, kidney, heart, and lung but also in non-vascularized islet cell transplantation for diabetes mellitus, even though transplantation has some limitations including graft failure or rejection, histocompatibility, the use of immunosuppressant, long-term survival of implanted organ. (4,5) Although advances in knowledge and technique continue to improve clinical outcomes of solid and tissue transplantation, clinical application to pituitary gland has been limited.(6)
  2. Page 2, line 51-53: We have deleted, “Pituitary transplantation, a theoretical curable treatment for hypopituitarism, may physiologically overcome these serious complications.”
  3. Page 2, line 61-64: We have added, “However, further investigations regarding pituitary transplantation for functional restoration of pituitary hormone rarely performed since 1998. Pituitary transplantation can be a theoretical curable method for hypopituitarism to physiologically overcome side effects of long-term hormonal replacement therapy.”
  4. We have added a reference and revised other references accordingly.
    1. Shapiro, A.M.J.; Pokrywczynska, M.; Ricordi, C. Clinical pancreatic islet transplantation. Nature Reviews Endocrinology 2017, 13, 268-277

Comment #2

The results presented are limited to measuring hormones, and immunohistochemical sections. Without other references.

Response #2

We thank the reviewer for their comment. As per your recommendation, we have described about the results of hormone and immunohistochemistry in more detail. However, comparative results of hormone and immunohistochemical stain between our study and previous studies were already described in the 2nd paragraph of the “Discussion” section with references. Therefore, we did not insert additional references in the “Results” section. All changes are highlighted in red.

  1. Page 5, line 199-202: We have revised the sentence to, “All rats after receiving pituitary transplantation showed rapid weight gain until 1 week (246.4±8g) compared to baseline (167±7.9g) and gradually gained weight until 11 weeks (295±4.5g at 11 weeks) after transplantation.”
  2. Page 5, line 210-213: We have added, “Mean serum GH level was higher in transplantation group until 11 weeks, but there was no significant difference from 3 weeks (7.715±9.923 versus 0.696±0.693ng/ml, p=0.152) after transplantation. The statistical difference of serum prolactin level was maintained until 7 weeks (4.765±0.544 versus 0ng/ml, p<0.001) after transplantation.
  3. Page 5, line 201-217: Detailed hormone levels were inserted in the second paragraph (3.2.) of the “Results” section. All changes are highlighted in red.
  4. Page 7, line 251-252: We have added, “Normal glandular structures with normal number of acidophils and basophils and without ischemic foci were seen in anterior hypophysis.”

Comment #3

The conclusions are very summarized and little elaborated and, the same happens with the discussion.

The references, although it is based on old works, should refer to more current publications.

Response #3

Thank you for your comments. As per your recommendation, we have revised the “Conclusion” and “Discussion” sections in more detail. Moreover, we agree with you that the references included in this study were somewhat old fashion. However, experimental and clinical studies for pituitary transplantation have been rarely performed since 1998 and thus, current publications regarding pituitary transplantation for hormone restoration has been extremely limited. Unfortunately, we cannot find representative current publications in spite of extensive search for this topic. Instead, we have added more recent 5 references regarding the disadvantage of pituitary transplantation, islet transplantation, and cell transplantation. All changes are highlighted in red.

  1. Page 2, line 49-51: We have added, “Although advances in knowledge and technique continue to improve clinical outcomes of solid and tissue transplantation, clinical application to pituitary gland has been limited.(6)”
  2. Page 8, line 281-285: We have added, “However, other hormones (TSH, GH, and prolactin) rapidly decreased at 2-4 weeks after hypophysectomy and this might be due to the functional decline of the remaining pituitary tissue over time. Therefore, we performed transplantation at 3 weeks after hypophysectomy to minimize the influence of remaining pituitary tissue on the restoration of hormones after pituitary implantation.”
  3. Page 8, line 315-316: We have added, “In addition, this induced hyperprolactinemia can lead to several complications. (7,8)”
  4. Page 9, line 364-367: We have added, “Moreover, because of the limitation of other sites including kidney capsule, ventricle and sellar floor as well as the omentum, further study regarding cell transplantation, such as stem cell or separated pituitary cells can be investigated. (30,31)”
  5. Page 10, line 385-387: We have added, “In this study, we found that TSH, GH, prolactin restored 1 week after transplantation while ACTH was not released from the graft. In addition, the structural preservation of implanted hypophysis could be assured by EM and IHC examinations Therefore,~~”
  6. We inserted 5 Additional references and revised other references accordingly.
    1. 5. Shapiro, A.M.J.; Pokrywczynska, M.; Ricordi, C. Clinical pancreatic islet transplantation. Nature Reviews Endocrinology 2017, 13, 268-277.
    2. 22. Asad, M.; Shewade, D.G.; Koumaravelou, K.; Abraham, B.K.; Balasinor, N.; Ramaswamy, S. Effect of hyperprolactinaemia as induced by pituitary homografts under kidney capsule on gastric and duodenal ulcers in rats. J Pharm Pharmacol 2001, 53, 1541-1547.
    3. 23. Moro, M.; Inada, Y.; Miyata, H.; Komatsu, H.; Kojima, M.; Tsujii, H. Effects of dopamine d2 receptor agonists in a pituitary transplantation-induced hyperprolactinaemia/anovulation model in rats. Clin Exp Pharmacol Physiol 2001, 28, 651-658.
    4. 30. Balyura, M.; Gelfgat, E.; Ehrhart-Bornstein, M.; Ludwig, B.; Gendler, Z.; Barkai, U.; Zimerman, B.; Rotem, A.; Block, N.L.; Schally, A.V., et al. Transplantation of bovine adrenocortical cells encapsulated in alginate. Proc Natl Acad Sci U S A 2015, 112, 2527-2532.
    5. 31. Lara-Velazquez, M.; Akinduro, O.O.; Reimer, R.; Woodmansee, W.W.; Quinones-Hinojosa, A. Stem cell therapy and its potential role in pituitary disorders. Curr Opin Endocrinol Diabetes Obes 2017, 24, 292-300.

Comment #5

The figures are in a very small format and are difficult to see.

Response #5

Thank you for your positive feedback. As per your recommendation, we have revised the Figures.

Comment #6

On the other hand, the pdf that has been sent to me to evaluate looks like a draft of the manuscript. I believe that the authors, should take care of the presentation so that the reviewer does not have difficulties to read the manuscript.

Response #6

Thank you for your comments. The manuscript that you previously received was our revised re-submitted manuscript. The manuscript could look like a draft because it might be tracking-on state. We have taken care of our manuscript according to the instruction for authors this time not to have difficulties for reviewers to read the manuscript.

Reviewer 3 Report

In this manuscript titled “Functional Restoration of Pituitary After Pituitary Allotransplantation into the hypophysectomized rats », the authors performed an hypophysectomy in 10 rats and grafted the removed pituitary in the omentum in four of them. They concluded that transplanted pituitary gland could survive in the omentum with partial restoration of anterior pituitary hormones.

Comments concerning the methods

The hypophysectomy, the pituitary graft in the omentum, the histological control (HIC and EM) of the graft at 3 days are well performed.  The two first techniques require a great surgical dexterity and the histology a good expertise. Would you mention the number of rats involved in this experiment? Especially what is the % of mortality and of uncomplete removal?

All the techniques are well illustrated, but the order of the figures must follow the text and the experimental procedure.  The figures 1, 5, and 6 are too big: Figure 1: only the diagram B, Figure 5: the curve of the weight could be removed, Figure 6: only five curves (TSH, GH, PRL, ACTH and weight). The tables of the values of each hormone and of the pituitary weight are not necessary. Their figures could be mentioned in the text. The video is not informative.

Comments concerning the conclusion

This study is an example of adapted procedure and good techniques to prove the success of hypophysectomy and ectopic allogenique transplantation of the pituitary. But is the omentum really the best location? In the discussion, you wrote that “the graft under the kidney capsule is known to be the best”. I agree. During forty years, I used this graft location for studying the pituitary tumor growth. When I grafted the normal pituitary, it remains well preserved, during more than seven months (unpublished results). Would you underline more clearly the limits of the graft in the omentum. The worst limitation is not the ectopic location without hypothalamic connection, but the fact that the pituitary was never found at 7 weeks and the secretions decreased quickly (2 weeks). You wrote “we believe that omentum can be a good candidate for pituitary transplantation. Would you compare your present results with the literature and study yourself the pituitary graft under the kidney capsule? In 2021, the interest of such allograft of the normal pituitary in the rat is limitated, especially when the pituitary is not found, at the end of the experiment. It may be useful to study the early effects of drugs, such as the oncologic immunotherapy which induce hypophysitis.

Author Response

Response to Reviewer 3 Comments

Comment #1

The hypophysectomy, the pituitary graft in the omentum, the histological control (HIC and EM) of the graft at 3 days are well performed.  The two first techniques require a great surgical dexterity and the histology a good expertise. Would you mention the number of rats involved in this experiment? Especially what is the % of mortality and of uncomplete removal?

Response #1

We thank the reviewer for their comment. We have previously reported the results of extracted pituitary gland preservation in 2019. At that time, we successfully performed parapharyngeal approach for pituitary extraction with a total of 19 rats and total number of rats used were 30. So, we have quite experience for parapharyngeal removal of rat pituitary gland. In this study, we used 10 rats for our 1st experiment (for identifying appropriate hypophysectomy). Among them, 6 were successfully removed and 3 were incompletely removed. One rat finally died. In 2nd experiment, another 8 rats were used and 4 were finally included in this study. 3 were incompletely removed and 1 ultimately died. Accordingly, a total of 18 rats were involved in this study. Mortality rate and incomplete removal rate were 11.1% and 33.3%, respectively.

Comment #2

All the techniques are well illustrated, but the order of the figures must follow the text and the experimental procedure.  The figures 1, 5, and 6 are too big: Figure 1: only the diagram B, Figure 5: the curve of the weight could be removed, Figure 6: only five curves (TSH, GH, PRL, ACTH and weight). The tables of the values of each hormone and of the pituitary weight are not necessary. Their figures could be mentioned in the text. The video is not informative.

Response #2

Thank you for your comment. As per your recommendation, we have revised our Figures and removed video clip. We have combined Figure 1 (A) and (B) to reduce the size. The graph for weight curve in Figure 5 and the table containing hormone levels (mean±SD) in Figure 6 were removed. Instead, we have mentioned the results in more detail in the text. All changes are highlight in red.

  1. Page 5, line 199-202: We have revised the sentence to, “All rats after receiving pituitary transplantation showed rapid weight gain until 1 week (246.4±8g) compared to baseline (167±7.9g) and gradually gained weight until 11 weeks (295±4.5g at 11 weeks) after transplantation.”
  2. Page 5, line 210-213: We have added, “Mean serum GH level was higher in transplantation group until 11 weeks, but there was no significant difference from 3 weeks (7.715±9.923 versus 0.696±0.693ng/ml, p=0.152) after transplantation. The statistical difference of serum prolactin level was maintained until 7 weeks (4.765±0.544 versus 0ng/ml, p<0.001) after transplantation.
  3. Page 5, line 201-217: Detailed hormone levels were inserted in the second paragraph (3.2.) of the “Results” section. All changes are highlighted in red.

Comment #3

This study is an example of adapted procedure and good techniques to prove the success of hypophysectomy and ectopic allogenique transplantation of the pituitary. But is the omentum really the best location? In the discussion, you wrote that “the graft under the kidney capsule is known to be the best”. I agree. During forty years, I used this graft location for studying the pituitary tumor growth. When I grafted the normal pituitary, it remains well preserved, during more than seven months (unpublished results). Would you underline more clearly the limits of the graft in the omentum. The worst limitation is not the ectopic location without hypothalamic connection, but the fact that the pituitary was never found at 7 weeks and the secretions decreased quickly (2 weeks). You wrote “we believe that omentum can be a good candidate for pituitary transplantation. Would you compare your present results with the literature and study yourself the pituitary graft under the kidney capsule? In 2021, the interest of such allograft of the normal pituitary in the rat is limitated, especially when the pituitary is not found, at the end of the experiment. It may be useful to study the early effects of drugs, such as the oncologic immunotherapy which induce hypophysitis.

Response #3

Thank you for your comment. We agree with you that the omental pouch may not an optimal site for pituitary transplantation. Also, we agree that the main drawback of omental pouch is that long-term survival cannot be possible in this study. Therefore, we tone down our suggestion regarding the omental pouch and highlight more clearly the limitation of omental transplantation. In addition, we found that the implanted pituitary gland can survive for more than 8 weeks under the kidney capsule, as similar to your comments. These mentions have added in the “Discussion” section of our revised manuscript. All changes are highlighted in red

  1. Page 9, line 340-341: We have changed the sentence to “Moreover, long-term survival of transplanted glands without revascularization is a main drawback of pituitary transplantation into omental pouch.”
  2. Page 9, line 343-344; We have inserted “In contrast, implanted pituitary gland beneath the kidney capsule can be preserved for more than 8 weeks.”
  3. Page 9, line 346-348: We have changed the sentence to “We thought that these results might provide the possibility of long-term survival of non-vascularized graft such as pituitary gland or islet cells inside the omental pouch.”
  4. Page 9, line 360-361: We have deleted “Although omental pouch has some limitations, we believe that this area can be a good candidate for pituitary transplantation due to some advantages.”
  5. Page 9, line 362-364: We have changed the sentence to “Because omental pouch has several limitations, additional investigations to find optimal site for transplantation and overcome these limitations would be needed.”

Round 2

Reviewer 2 Report

now it is OK

This manuscript is a resubmission of an earlier submission. The following is a list of the peer review reports and author responses from that submission.

Round 1

Reviewer 1 Report

In this paper the authors have transplanted pituitary tissue into the omentum of hypophysectomised rats to determine if this leads to replacement of the circulating hormones normally released by the pituitary. The authors report sustained increases in circulating TSH and some restoration of circulating GH. The study includes some interesting data, not least as I am surprised at the level of functional restoration was achieved, however there are several issues thta need addressing.

Major points:

  1. The data shown in figure 4 would indicate to me that there was substantial pituitary tissue remaining after hypophysectomy, albeit with a damaged hypothalamic connection. Given the short half-life of pituitary hormones (minutes), the persistent circulating ACTH even after 4 weeks, would suggest a substantial portion of the pituitary must still be present. In light of this, the restored  TSH and GH may be from the remaining pituitary tissue. Why this should increase after transplantation into the omentum is unclear but is a strong possibility.
  2. In the classical models using pituitary transplant prolactin is dramatically increased as a result of the loss of negative feedback from the hypothalamus. This would have serious health issues (not least infertility)- prolactin needs to be measured in these studies.
  3. A key feature of pituitary biology is its relationship with the hypothalamus and the specialised blood supply delivering a high concentration of hypothalamic factors directly from the median eminence. This is hardly mentioned in the manuscript. Restoration of physiological levels and pattern of pituitary hormones requires this hypothalamic input. This is hardly mentioned in the manuscript but would be a key question in improving on current replacement therapies by transplantation. This point at least requires discussion.
  4. Single time-point measurements of pulsatile hormones (eg GH and ACTH) are not meaningful. This requires multiple samples over a period of hours to discern the basal and pulsatile levels that differentially regulate physiology.

Minor points:

  1. Graphs in Figures 4 and 5 are missing a Y-axis label.
  2. Graphs in figure 6 should,show mean+/- SEM.

Reviewer 2 Report

The authors reported their findings of pituitary function recovery after grafting pituitary into the omentum pourch of the hypophysectomized rats. The experiments were well designed and the findings were properly interpreted. So the overall scientific merit was well qualified for Cells readers. I have several points for the authors to consider to include in their revision.

(1)Among the 3 pituitary hormones tested, ACTH was outstanding in terms of delayed depletion compared to TSH and GH following hypophysectomy. Is this likely due to the distribution/percentage of corticotroph cells in the rat pituitary so that the surgical disruption did not eradicate sufficiently? And the remaining corticotroph cells compensate the loss through HPA axis?

(2)Fig.7 examined the allotransplanted pituitary of 3 days with necrotic spots. Have the authors checked long-term grafted pituitary? Assuming absence of hypothalamic communicating signals, and limited blood supply from omentum pourch,  the grafted pituitary may not response normally to stress or physiologic challenges. Can the authors state on this?